



# Deciphering Organization of GOES–16 Green Cumulus, through the EOF Lens

Tom Dror[1], Mickaël D. Chekroun[1], Orit Altaratz[1], and Ilan Koren[1]

[1]Department of Earth and Planetary Sciences, Weizmann Institute of Science, Rehovot, Israel.

**Correspondence:** Ilan Koren (Ilan.koren@weizmann.ac.il)

**Abstract.**

A subset of continental shallow convective Cumulus (Cu) cloud fields were shown to have unique spatial properties and to form mostly over forests and vegetated areas, thus referred to as *green Cu*. Green Cu fields are known to form organized mesoscale patterns, yet the underlying mechanisms as well as the time variability of these patterns are still lacking understand-
ing. Here, we characterize the organization of green Cu in space and time, by using data driven organization metrics, and by applying an Empirical Orthogonal Function (EOF) analysis to a high-resolution GOES–16 dataset. We extract, quantify and reveal modes of organization present in a green Cu field, during the course of a day. The EOF decomposition is able to show the field's key organization features such as cloud streets, and it also delineates the less visible ones, as the propagation of gravity waves (GW), and the emergence of a highly organized grid on a spatial scale of hundreds of kilometers, over a time
period that scales with the field's lifetime. Using cloud fields that were reconstructed from different subgroups of modes, we quantify the cloud street's wavelength and aspect ratio, as well as the GW dominant period.

## 1 Introduction

The emergence of organized patterns in cloud fields is ubiquitous, observed throughout different cloud types around the world, across a wide range of scales. Shallow cumulus (Cu) cover large areas over the oceans and continents (Norris, 1998; Bony
et al., 2004), reflect part of the incoming solar radiation but have minor influence on the outgoing longwave radiation (OLR) (Turner et al., 2007; Berg et al., 2011), thus they contribute to a net cooling effect of the planet (Boucher et al., 2013). However, despite their great influence on the Earth's radiation budget and the overall climate sensitivity, they still account for much of the uncertainty associated with cloud feedback (Bony, 2005; Webb et al., 2006).

It has long been recognized that shallow Cu are organized on the mesoscale $(20 - 2,000\,km)$ (Agee et al., 1973). Shallow Cu
fields exhibit a variety of patterns such as cloud streets (Brown, 1980), clusters (Zhu et al., 1992; Heus and Seifert, 2013), skeletal networks, or mesoscale arcs (Stevens et al., 2019). Such organized patterns of a cloud field result often from the interaction between the internal nonlinear dynamics (self-organization) and the external forcings (Klitch et al., 1985). Several mechanisms for self-organization have been proposed; from gravity waves (GW) (Atkinson and Wu Zhang, 1996; Dagan et al., 2018) to interactions between water vapor and radiations (Wing and Emanuel, 2014), to precipitation-induced cold pools (Xue
et al., 2008; Seifert and Heus, 2013). External forcing may result from e.g. changes in topography, land cover (Rabin and





Martin, 1996), or soil moisture (Ray et al., 2003). Related processes impact the spatial partitioning of updrafts and downdrafts, and therefore affect the overall structural properties of the field's patterns. These properties determine the clouds' location and their size distribution within the field (Seifert and Heus, 2013), and play a key role in determining the radiative effects (Tobin et al., 2012).


Continental Shallow Cu, often forming during the summer season (Zhang and Klein, 2013) or dry season in the tropics, are observed in a variety of locations, from low- to mid- and high-latitudes, and are preferably formed over forests and vegetated areas, therefore referred to as *green Cu* (Dror et al., 2020). These continental clouds in general, and specifically their organization are by far less studied than their maritime counterpart, i.e., trade Cu. The level and type of organization of con-

tinental shallow Cu were shown to often exhibit regular (grid-like) pattern and cloud streets (Nair et al., 1998; Dror et al., 2020), the former being a special case of the latter. Complimentary, cloud streets are roll vortices that are often defined as quasi two-dimensional organized large eddies whose wavelength is scaled by the convective boundary layer (CBL) depth with orientation along with the mean CBL wind direction (Etling and Brown, 1993). Within this dynamical environment, the clouds form above the updraft branch of the roll circulation, while cloud-free areas are associated with the drier air in the downdraft

branch (Weckwerth et al., 1997). Cloud streets result from a combination of steady wind shear and buoyant parcels rising from the surface (Brown, 1980), and were recently shown to also be affected by 3D radiation effects, mostly due to cloud's shadow (Jakub and Mayer, 2017).

Yet, the understanding of the aforementioned patterns and their time-variability in shallow Cu fields along with the understanding of their potential role in low cloud feedback, remain limited (Vial et al., 2017; Nuijens and Siebesma, 2019). In that respect,

high-end simulations are not concurring favorably as they fail in reproducing such features either in cloud-resolving models or in large eddy simulation models. As a consequence shallow Cu mesoscale organization is not properly represented in global circulation models (GCMs).

Within this context, we propose in this work to extract and quantify modes of organized convection present in green Cu, by application of Empirical Orthogonal Function (EOF) decomposition (Fukuoka, 1951; Lorenz, 1956), also known as Principal

Component Analysis (Jolliffe, 2002), to high-resolution satellite data. EOF analysis is among the simplest approaches for data decomposition of spatio-temporal fields and is widely used in atmospheric science but seems to have been underexploited for analyzing cloud fields. The purpose of this study is to bridge this gap and to show the usefulness of EOF analysis in extracting organizational structures from a complex cloud field. Given a spatio-temporal signal, recall that an EOF analysis provides its variance decomposition which results into a data-driven separation of variables: spatial patterns ranked in terms of their

variance contributions (EOF modes) and principal components (PCs) characterizing the time-variability of these modes; see Eq. (1).

EOF decomposition has been applied to various atmospheric, oceanic, or climate fields for various purposes such as: exploratory data analysis, dynamical mode reduction, pattern extraction and data-driven stochastic modeling. Among the typical observational fields and oscillations examined we can mention the sea-level pressure (SLP) to extract individual modes of

variability, such as the Arctic Oscillation (see e.g., Thompson and Wallace (1998)), the sea surface temperature (SST) and the





El Niño-Southern Oscillation (Penland and Magorian, 1993; Penland and Sardeshmukh, 1995), and the Atlantic Multidecadal Oscillation (Messié and Chavez, 2011). Among the issues encountered in practice, it is known that the orthogonality constraint inherent to the EOF modes, makes the physical interpretation of their patterns sometimes non-trivial (Monahan et al., 2009), and various extensions of EOF decomposition have been proposed to remediate to such shortcomings. Such extensions include

rotated EOF (ROEF) and the like (Horel, 1981; Richman, 1981, 1986; Cheng et al., 1995; DelSole and Tippett, 2009) applied e.g., to SST (Kawamura, 1994; Mestas-Nuñez and Enfield, 1999) and SLP records (Hannachi et al., 2006); extended EOFs (EEOFs) (Weare and Nasstrom, 1982) applied e.g. to OLR to analyze large-scale organized deep, tropical convection (Roundy and Schreck III, 2009); and other multivariate spectral analysis methods which help reduce undesirable mixture effects among frequencies (Groth and Ghil, 2011; Chekroun and Kondrashov, 2017).

As shown below, a standard EOF analysis on our dataset does not suffer from such shortcomings, and physical interpretations can be unambiguously drawn from the resulting EOF modes and their time-variability. What makes this study distinct from the aforementioned works, is tied to the scales analyzed here. Typically, an EOF analysis is performed on coarse, synoptic- to global-scale observations over time scales that may span decades and even up to centuries in the case of model simulations (Chen et al., 2016). In contrast, we apply the EOF method over a short time window, during the course of a day, to analyze

finer scales patterns contained in high-resolution — in both space and time — satellite observations from GOES–16 (Schmit et al., 2017) over an area and time frame dominated by cloud streets patterns exhibited by a green Cu field over continental US (CONUS). Decomposing the original complex and nonlinear field into elemental structures that are interpretable from a physical viewpoint, is a challenging albeit an important task. The goal of the EOF analysis presented here is twofold: (i) to advance understanding about the physical mechanisms at play in the generation of mesoscale patterns observed in green Cu

fields, and (ii) to provide tools for describing cloud field organization, which can be used to compare between simulations and observations for improved representation of mesoscale dynamics in high-resolution numerical models. As discussed below, and beyond the reduction of the field's dimensionality, the EOF analysis is not only able to capture these dominant patterns, but allows for exhibiting the presence of GW, delineating thus several processes underlying the multi-scale variability of the field.

## 85   2   Methods

### 2.1   GOES–16 ABI Data and Data Preparation

The Advanced Baseline Imager (ABI) on board GOES–16 is a state-of-the-art 16-band radiometer, providing four times the spatial resolution, and more than five times faster temporal coverage than the former GOES system (Schmit et al., 2017). We use the ABI's level 1B "Red" band (channel 2, $0.64\,\mu m$) radiance, which has the finest resolution (0.5 km) of all ABI bands, over

CONUS (temporal resolution of five minutes). The "Red" band detects reflected visible solar radiation, and its high-resolution makes it ideal for exploring green Cu during daytime. To prepare these data for EOF analysis, we converted the radiance values to reflectance following Schmit et al. (2010), applied a simple gamma correction to adjust and brighten the images (see Text S1), and reprojected the data from their native geostationary projection into a geographic (latitude–longitude) one. We focus





on a vast region of interest (ROI), located between $30°N - 36°N$ and $95°W - 80°W$, on a day that features a typical case of
daytime, locally formed shallow convection, from late-morning (10:47 Eastern standard time [EST]) to the afternoon (18:47
EST) of August 22nd, 2018, corresponding to a field ($F$) comprised of 97, 2D snapshots of corrected reflectance over 1335
latitude grid points ($\theta$) and 3339 longitude grid points ($\phi$), for a total of eight hours. The ROI spans two different time zones,
such that the local time at the eastern and western parts are four and five hours behind coordinated universal time (UTC),
respectively. Note that we present the time in EST throughout the paper.

## 2.2 Metrics of Organization

To derive the data driven organization metrics of Sec. 3, a threshold of 0.1 was applied to the reflectance values to roughly
discriminate between cloudy and non-cloudy pixels. Cloud objects were then identified by their centroids, using a pixel con-
nectivity of four. Metrics such as cloud fraction (CF), number of clouds (N) and the observed nearest neighbour cumulative
density functions (NNCDF) were calculated. The observed NNCDF is plotted against the Poisson NNCDF. The latter repre-
sents a field randomly distributed (with NNCDF which is given by the Weibull distribution; Weger et al. (1992)). Finally, the
organization index ($I_{org}$; Tompkins and Semie (2017)) is computed by integrating the area below the observed NNCDF. A
randomly distributed field would result in an $I_{org} = 0.5$, while any value lower (higher) than that corresponds to a regular,
grid-like (clustered) organization.

## 2.3 EOF Decomposition

To perform the EOF decomposition, we first concatenate the two spatial dimensions, latitude $\theta$ and longitude $\phi$, and transform
the cloud field $F$ into a space–time scalar field $X(t, \boldsymbol{x})$, representing the value of the corrected reflectance at time $t$ and at the
concatenated spatial variable $\boldsymbol{x}$. Second, the anomaly field $X'(t, \boldsymbol{x})$ is formed by subtracting the time-averaged field $\overline{X}(\boldsymbol{x})$ to
the original field, namely

$$X'(t, \boldsymbol{x}) = X(t, \boldsymbol{x}) - \overline{X}(\boldsymbol{x})$$

The spatial covariance matrix $\boldsymbol{C_X}$ of $X'$ is then estimated and the first 20 EOFs, $E_k$, $k = 1, \cdots, 20$ are computed. The latter
are the 20 first leading eigenvectors of $\boldsymbol{C_X}$ after sorting the eigenvalues, $\lambda_k$, in decreasing order. The statistical significance
of these dominant modes is assessed by their standard sampling error estimate, $\lambda_k \sqrt{(2/N)}$, where $N$ denotes the sample size
(North et al., 1982); here $N = 97$. The decomposition of the anomaly field $X'(t, \boldsymbol{x})$ is then obtained as follows:

$$X'(t, \boldsymbol{x}) = \sum_{k=1}^{M} c_k(t) E_k(\boldsymbol{x}), \tag{1}$$

where $c_k$ is the $k$-th PC, and $M$ is the number of EOF modes. Finally, the reconstructed field is obtained by adding back the
mean field $\overline{X}(\boldsymbol{x})$, and unpacking the concatenated variable, $\boldsymbol{x}$, back to the original spatial variables $\theta$ and $\phi$. Here, the first 20
EOFs account for 65.6% of the variance.



## 3  Data Driven Organization of Green Cu

As mentioned above, we focus on a mesoscale green Cu field over CONUS during August 22nd, 2018, and apply the EOF
analysis of Sec. 2.3 to ABI "Red" visible band corrected reflectance. Figure 1 shows the diurnal evolution of the field with
snapshots taken at 11:32, 13:47, 16:02, and 18:17 EST. The field starts to develop at late morning (Fig. 1-(a)) as the surface

125  warms up and thermal convection begins (Stull, 1985). The field is at its peak in terms of reflectance from around noon to
early afternoon (Fig. 1-(b,c)), and start dissipating in late-afternoon (Fig. 1-(d)), as the surface cools down and surface fluxes
die out. During daytime, green Cu emerge and organize in a distinguishable fashion to form cloud streets, generally oriented
from north to south (turn into a west–east orientation at the eastern part of the ROI, see Fig. 1-(b,c)) which are maintained
throughout the field's lifetime; see video in SI. Further evidence of the cloud streets is demonstrated by examining the patterns

130  of the time–averaged field, $\overline{X}(\boldsymbol{x})$ (see details in Sec. 2.3), shown in Fig. 2. The cloud streets are still visible in $\overline{X}(\boldsymbol{x})$ as linear
features that partition the field to stripes of cloudiness and clear skies, indicative thus of a slow drift of these patterns during
their lifetime.

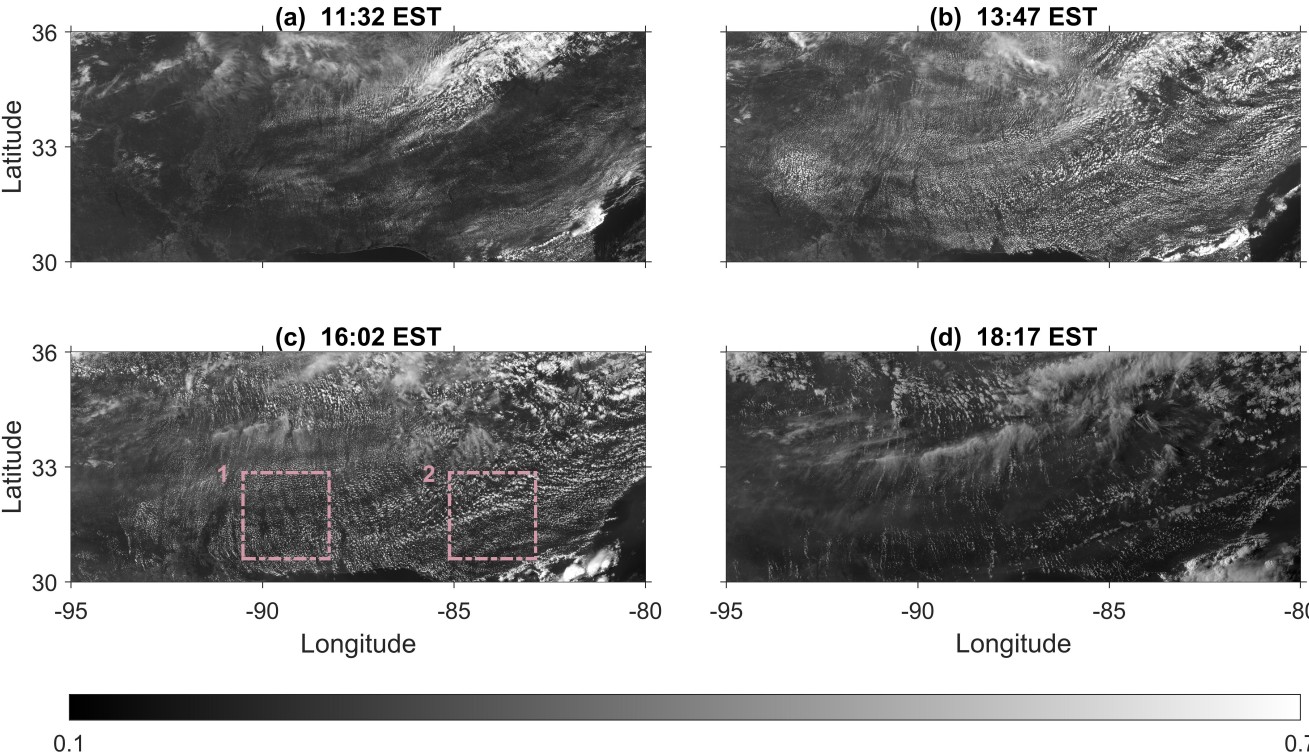

**Figure 1.** Corrected reflectance field of the diurnal cloud evolution as obtained from GOES–16 ABI's "Red" visible band (channel 2;
$0.64\,\mu m$). Snapshots taken at 11:32 (a), 13:47 (b), 16:02 (c), and 18:17 (d) EST. Dashed boxes in (c) represent two subdomains analysed in
Fig. 3.





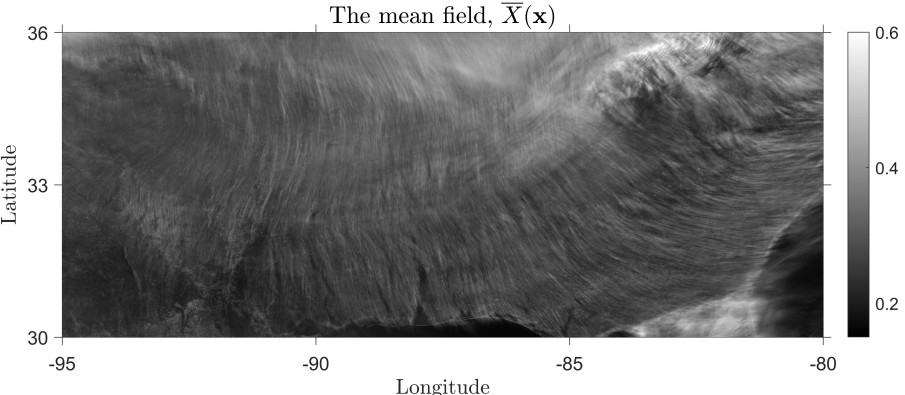

**Figure 2.** The time–averaged corrected reflectance field $\overline{X}(\boldsymbol{x})$. Note the clear stripes in $\overline{X}(\boldsymbol{x})$, marking the cloud streets along the mean CBL wind direction.

We focus on two subdomains (see boxes in Fig. 1(c)) and explore their organization (Fig. 3). While both of the subdomains exhibit a cloud street pattern, subdomain–2 also features the presence of GW in the afternoon (indicated by a dashed line in Fig. 3(b)). GW are frequently present in the atmosphere, they often occur over flat terrain and are ubiquitous over shallow Cu fields. In fact, the most intense and best organized GW were reported over cloud streets (Kuettner et al., 1987). The latter, also known as convective GW, are generated internally by the field with shallow Cu and/or the cloud streets acting as convective obstacles to the mean horizontal flow (Jaeckisch, 1972; Simpson, 1983; Melfi and Palm, 2012). Once excited, the GW can act as a feedback mechanism to reorganize the convection, implying that the stable layer overlying the inversion plays an important role in determining the field's organization (Clark et al., 1986). In Fig. 3 we examine the effect of GW, that may be challenging to observe with the naked eye, on the organization metrics of Sec. 2.2. The GW evident in subdomain–2 result in larger-sized clouds, which is manifested in increased CF, and decreased N during the afternoon (15:02–18:02, Fig. 3(c)). By plotting the NNCDFs of the clouds' centroids in the two subdomains, against the one of a randomly distributed cloud field given by the Poisson NNCDF (Weger et al., 1992) (Fig. 3(d)), we show that the two subdomains deviate from a random organization ($I_{org} < 0.5$, see sec. 2.2) and exhibit a regular (grid-like) pattern, which is a high level of organization. This type of organization is typical for green Cu fields (Dror et al., 2020) and was reported in several continental areas, e.g., the Amazon basin (Da Silva et al., 2011; Heiblum et al., 2014), the Southern Great Plains (Hinkelman and Evans), and Northern Germany (Müller et al., 1985). However, due to the presence of GW, the curve of the observed NNCDF of subdomain–2 lies closer to the diagonal, and leans more towards clustered organization, resulting in a larger $I_{org}$ comparing to that of subdomain–1 ($I_{org} = 0.43$ and $I_{org} = 0.34$, respectively).

We note that even-though we focus on the green Cu field, that covers most of the ROI during the day, there are other types of clouds that appear in the ROI; deeper orographic warm clouds in the upper-right corner (south edge of the Appalachian Mountains), mostly during noon-early afternoon, cirrus clouds at the north and central parts of the ROI, moving southward along the day (located above the Cu clouds), shallow early-morning clouds that dissipate ~11:30 EST norther to Florida, and

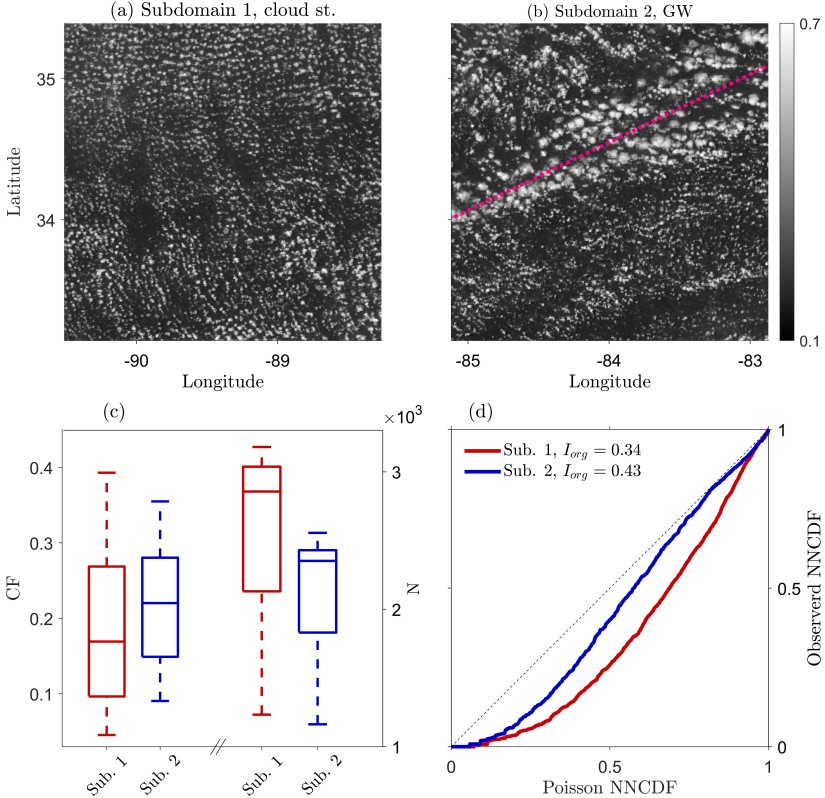

**Figure 3.** Snapshots of the two subdomains marked in Fig. 1-(c) at 16:07 EST (a). Dashed magenta line in (b) shows the front of the GW. (c) Boxplots of the cloud fraction (CF) and the number of clouds (N) for subdomain–1 (red), and subdomain–2 (blue) between 15:02–18:02 EST. (d) Observed NNCDF against Poisson NNCDF for the two subdomains (color specification follows the one in (c)) at 16:07 EST. Diagonal dashed line shows a randomly distributed cloud field, while deviations above (below) the diagonal indicate a tendency toward clustering (regularity). The organization index ($I_{org}$; Tompkins and Semie (2017)) of each subdomain is given in the top-left corner.

deep convective clouds at the lower-right corner of the ROI that develop along the day (see SI for more details regarding the cloud classification). However, the corrected reflectance field is clearly dominated by green Cu during the inspected time-frame.

## 4 Organization in Space and Time: EOF

We further examine the clouds field organization by conducting an EOF decomposition to the corrected reflectance images. Figure 4 shows the spectrum of the covariance matrix $C_X$ (see Sec. 2.3), and the field's key spatial features obtained by the EOF decomposition such as exhibited by the first three EOF modes; we refer to Figs. S5–S7 for the other EOF modes. The spectrum of the covariance matrix $C_X$, shown in Fig. 4-(a), is composed of the eigenvalues $\lambda_1, \cdots, \lambda_{20}$ which informs on the distribution of energy and on the separation/degeneracy of the EOF patterns. The corresponding PCs are shown in Fig. 5-(a).



The leading mode (EOF1, 31.8%) is nondegenerate (i.e., $\lambda_1$ is well separated from the rest the $\lambda_k$s) and explains more than

twice the variance than any other mode. Although the EOF decomposition is performed on the anomaly field (see Sec. 2.3), correlations between EOF1 and the mean-field still remain (compare Fig. 2 and Fig. 4-(b)), showing these patterns constitute an important part of the field's variability. Thus, EOF1 displays the streets pattern of the green Cu field (Fig. 4-(b)) while PC1 (Fig. 5a) shows a diurnal cycle manifestation whose magnitude peaks around local noon (in absolute value) coinciding with the evolution of green Cu. This is somehow not surprising as these clouds are closely linked to thermal convection, thus, their

properties are tightly tied to the diurnal cycle of the surface fluxes (Stull, 1985; Zhu and Albrecht, 2002). They usually form around mid- or late-morning and dissipate after sunset (Zhu, 2003; Berg and Kassianov, 2008; Zhang and Klein, 2013). The second mode (EOF2, 14.8%), also nondegenerate, is mostly pronounced at the north-eastern and south-eastern parts of the ROI, displaying the general location of the orographic, shallow early-morning clouds, and deep clouds marked in Fig. 4-(c) as closed areas delimited by dash-dot, dashed and dotted curves, respectively (further evidences of these aspects are provided in

the SI). PC2 is also dominated by a diurnal cycle, but mostly pronounced in morning and late afternoon, i.e., before green Cu emerge, and after they disappear. The third mode (EOF3, 7.4%) is marginally degenerate (depending on whether considering the errorbars or not), and displays the structures of cirrus clouds at the northern part of the ROI (area enclosed by dotted curves in Fig. 4-(d)). Furthermore, on the south-eastern corner, we observe a blue-red contrast in patterns indicative that EOF3 captures also some features of the deep-convective clouds more visible in EOF2; see also Fig. S5. An inspection of Fig. S5 reveals

patterns that seem to propagate transversely to the cloud streets, as indicated by dashed lines therein.

     We turn now to a finer analysis of such patterns and EOF modes in general. The goal is to gain further understanding about the structures and scales exhibited by the EOFs, by analyzing the time-variability of the corresponding PCs, providing the EOF's amplitude. Generally, in the case of a forward cascade of energy dissipation, a decaying energy distribution such as

shown in Fig. 4-(a) is thought to decay as a function of the spatial scale, with the large-scale patterns that tend to capture most of the variance while evolving in time at a typical low-frequency for the multivariate signal analyzed. But a word of caution must be mentioned here. This scenario of energy decay as a function of well-separated scales is indeed not always satisfied, and EOF modes exhibit sometimes a mixture of spatial scales, leading to various degree of harmfulness for the analysis. Fortunately, for the dataset analyzed here, this phenomenon is not very pronounced. Indeed, only "an echo" of the cloud street's patterns

mostly contained in EOF1, subsists in the higher modes (see Fig. 4-(c,d) and Fig. S5) which benefits thus to the analysis and interpretation.

  As a result, by going down into the energy spectrum, the PCs move from low- to high- and higher-frequencies. Low (resp. high) frequency translates to EOFs dominated by large (resp. small) scale patterns, and intermediate frequency to a mixture of these scale; see Figs. S5–S7. Grouping the PCs accordingly, the PC1–PC2 pair corresponds mainly to a low, diurnal-like time

variability, PC3–PC9 all share an intermediate time variability, while PC10–PC20 are not only of higher frequency, but also oscillate mainly during the time-window over which PC1 peaks, i.e., when green Cu are mostly pronounced. Among the PCs exhibiting an intermediate range of time-variability (PC3–PC9), PC5 and PC6 are distinguished. The latter indeed reveal as forming an oscillatory pair, manifested by a phase shift between them in the time-domain and by a nearly-periodic behavior





expressed by a nearly-closed curve into the reduced state-space spanned by EOF5 and EOF6; see Fig. 5. This pair of EOFs

correspond to a $1.5\,hr$ near-period oscillation interpreted as the fingerprint of GW travelling throughout the field (see e.g., Fig. 3-(b)). GW may propagate both horizontally and vertically above the inversion, and were shown to have horizontal wavelengths of a few dozens of kilometers (Stull, 1976; Lane and Reeder, 2001; Lane, 2015) and to vertically extend throughout the whole troposphere (Clark et al., 1986; Kuettner et al., 1987; Hauf and Clark, 1989). Here we observe horizontally propagating GW, with wavelength of $\sim 130\,km$ as estimated from the stripe patterns in EOF5 and EOF6; see Text S1 and Fig. S5. A

complimentary analysis which consists of inspecting the time evolution of the field itself and its EOF reconstruction across a transection (red line in Fig. 6-(a,b)) supports this interpretation as discussed next.

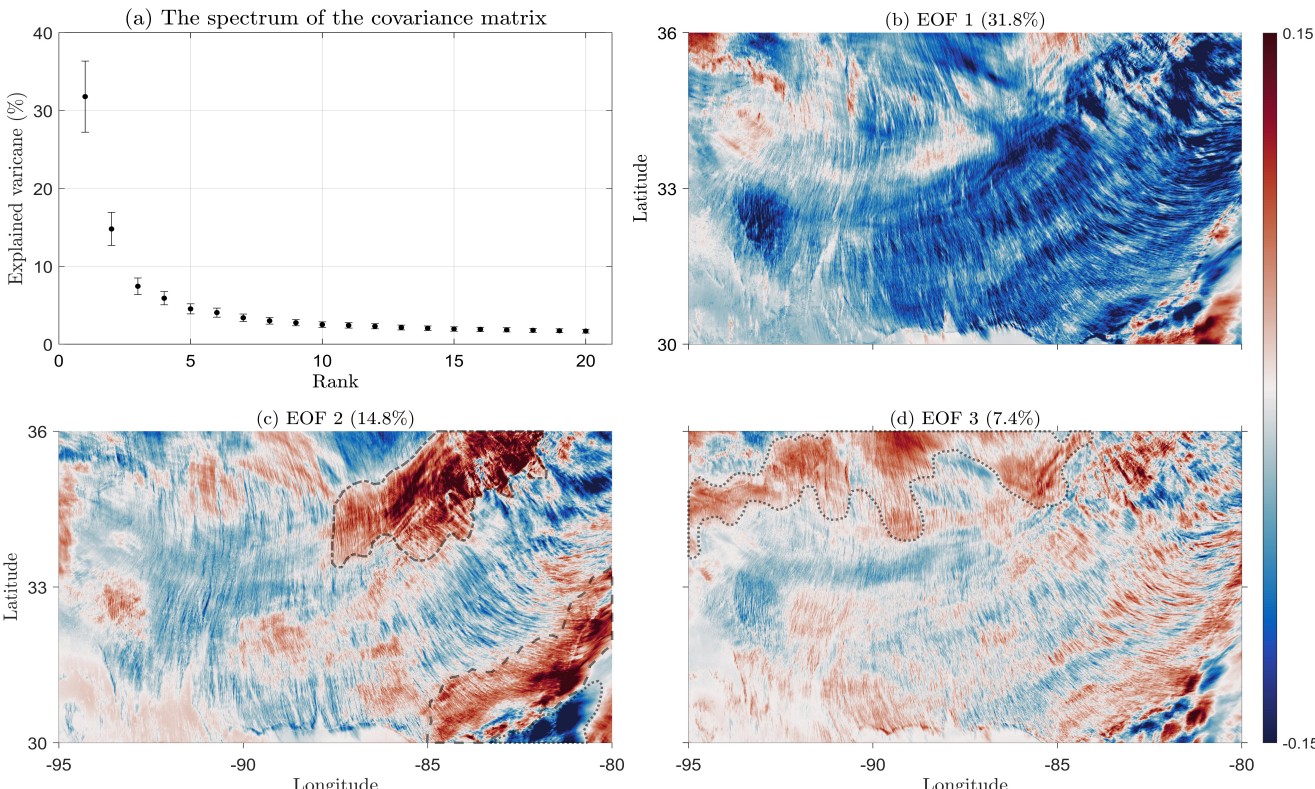

**Figure 4.** (a) Spectrum (%) of the covariance matrix. Errorbars calculation followed (North et al., 1982). (b–d) The three leading EOF modes, units are arbitrary (the variance explained by each mode is noted at the title of each panel). Structures related to orographic, morning and deep clouds are enclosed in dash-dote, dashed and dotted lines in (c), respectively. Structure related to cirrus clouds is enclosed in dotted line in (d).

To better examine the presence of GW in our green Cu field, we analyse the time-evolution of the field across the transection orthogonal to the cloud streets, marked by the red line in Fig. 6-(a,b). The corresponding Hovmöller diagrams of the field's diurnal evolution are shown for the corrected reflectance data, and for the reconstructed fields obtained by adding to the



**Figure 5.** Time series of all the 20 PCs, PC5 and PC6 are marked in black and gray, respectively (a). Time series of PC5 and PC6 (b), and phase diagram of PC5 *versus* PC6 (c).

mean-field $\overline{X}(x)$, the sum of the PCs multiplied by their corresponding EOFs as in the right-hand-side of Eq. (1), but for three groups of modes as follows: (i) the two leading ones (PCs 1–2 × EOFs 1–2), (ii) the intermediate ones (PCs 3–9 × EOFs 3–9), and (iii) the high-frequency ones (PCs 10–20 × EOFs 10–20). As mentioned above, the grouping of these modes is made according to their spatial degeneracy (Fig. 4-(a)) on one hand, and their temporal-frequency (Fig. 5-(a)), on the other. Recall that



EOF modes 1–2 are both nondegenerate and feature a low, diurnal frequency. EOF modes 3–9 are marginally degenerate, still
featuring substantial vertical change in Fig. 4-(a), and share an intermediate temporal frequency. EOF modes 10–20, however,
are degenerate and belong to the flat part of the spectrum of the covariance matrix.

A snapshot (16:07 EST) of the field versus the reconstructed field (using the 20 EOF modes) is shown in Fig. 6-(a,b), respec-
tively. The Hovmöller diagram obtained from the raw data is noisy (Fig. 6-(c)), as it represents the superposition of several
phenomena of different temporal- and spatial-scales. While the diurnal cycle of green Cu field is captured, with high reflectance
values that start to appear in late morning-time (∼11:00 EST), and disappear in the afternoon (∼18:00 EST). The prominent
feature of the field's organization, i.e., the cloud streets, are only somewhat captured and appear as elongated discontinuous
vertical features of high reflectance, composed of several reflectance blobs that represent the individual clouds that are ad-
vected through the transection. However, by decomposing the original, highly complex field, the diagrams of the reconstructed
fields reveal a much clearer picture, and allow us to extract important parameters of the field. The cloud streets in Fig. 6-(d)
appear continuous, more pronounced and well-defined, and the spacing between them is also clearer (vertical features of low
reflectance). Based on Fig. 6-(d), we estimate the street's wavelength as well as their aspect ratio (i.e., the street's wavelength
divided by the CBL depth; Young et al. (2002), see Text S1) to vary between $3 - 10.5\,km$. These values agree well with the
range reported for continental cloud streets (Etling and Brown, 1993; Young et al., 2002; Da Silva et al., 2011). GW, hardly
observable in Fig. 6-(c) appear to be more distinguishable in Fig. 6-(e) as nearly-horizontal stripes of high reflectance. Within
this representation, at least 2–3 waves appear during the day, with a dominant period of $\sim1.5\,hr$, in agreement with reported
timescales about GW (Tsuda, 2014; Nappo, 2013). Finally, EOF10–EOF20 are able to capture individual clouds that form and
dissipate as "pearls on a string" along the cloud streets throughout the day (see vertical oscillations along each cloud street in
Fig. 6-(f)). These higher frequency modes also indicate that green Cu in adjacent cloud streets, that may be distant apart by
tens to hundreds of kilometers, tend to form and dissipate in phase to form an organized structure also on the axis normal to
the cloud streets (see horizontal oscillations in Fig. 6-(f)), shaping thus an immense, highly organized grid of clouds, that lasts
throughout the field's lifetime.

## 5   Summary and Discussion

In this article, we proposed a new approach combining GOES–16 ABI's high-resolution corrected reflectance data, organiza-
tion metrics and an EOF analysis to investigate and characterize the mesoscale patterns obtained by a vast shallow Cu field
over CONUS, during August 22nd 2018. Organized shallow Cu, referred to also as green Cu, start to form around late-morning
in response to daytime surface heating, reach their peak (in terms of cloud cover) from noon to early-afternoon, and dissipate
in the afternoon (Fig. 1). Cloud streets are evident throughout the whole domain, their axis aligned along the mean advection
direction, i.e., mostly in the north–south direction (northerlies). The cloud streets are sustained throughout the field's lifetime,
and maintain approximately fixed positions, which is apparent in the time-averaged field (Fig. 2). By focusing on two sub-
domains we show that GW, that are orthogonal to the cloud streets, travel through the field and affect the organization by
clustering the clouds, thus making them larger and fewer, and that the clouds' organization deviates from randomness to a





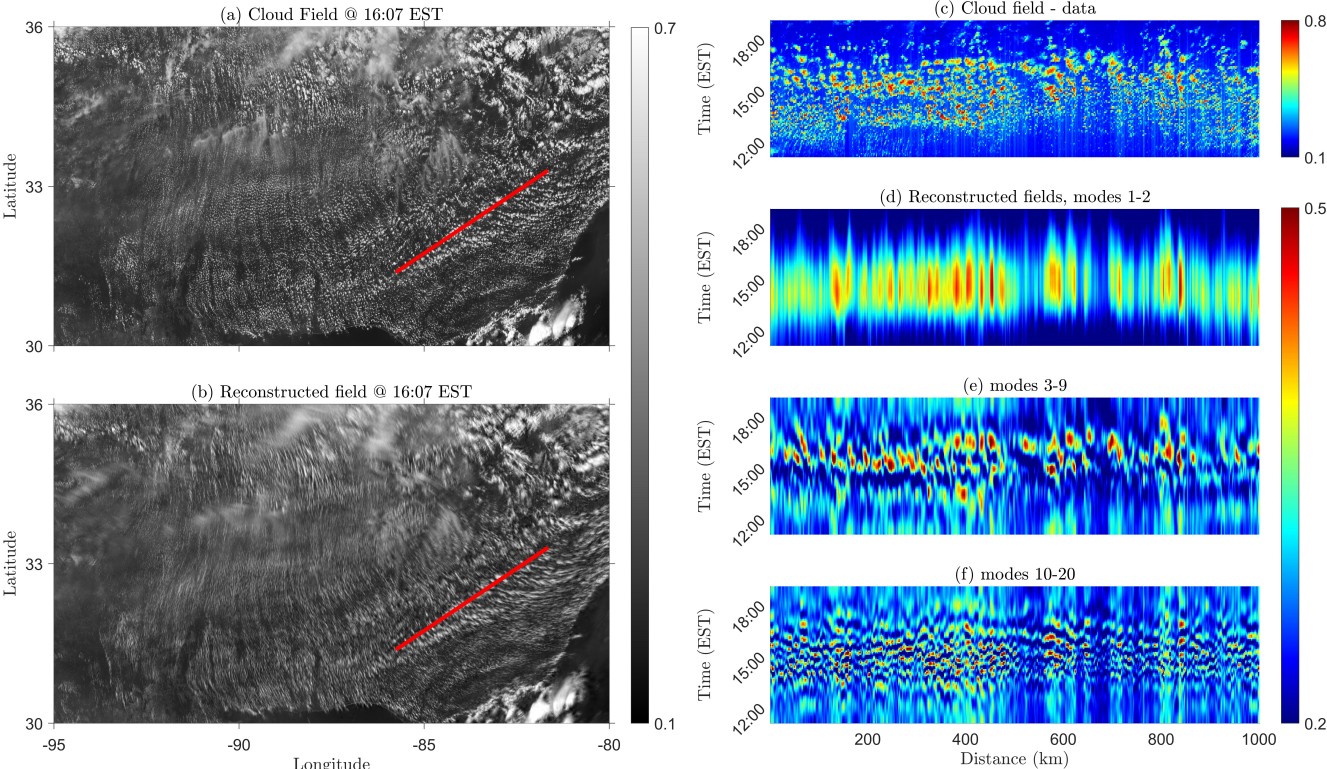

**Figure 6.** Snapshot of the cloud field (a) and the field reconstructed using all 20 modes (b) at 16:07 EST. (c) Corrected reflectance as a function of time along the transaction marked in red in (a,b). Same as in (c) but for the field reconstructed from modes 1–2 (d), modes 3–9 (e), and modes 10–20 (f).

grid-like organization type (Fig. 3). The three leading modes of the EOF decomposition (Fig. 4-(b–d)) reveal the field's most dominant spatial features, and the spectrum of the covariance matrix (Fig. 4-(a)) shows the degeneracy of the leading EOF modes. The structure of the cloud streets, formed by the green Cu, is well-captured by the leading EOF1, while (the nonde-

generate) EOF2 relates to structures of other types of clouds that exist in the ROI, namely orographic, shallow early–morning clouds and deep clouds. EOF3 is marginally degenerate, and therefore harder to relate to a specific physical phenomena, it contains a mixture of scales and structures, such as patterns of cirrus, deep clouds and GW. Over the time frame analyzed here, the time-variability associated with the EOF modes is shown in Fig. 5-(a). The related PCs exhibit an organized set of time scales ranging from from low- to intermediate- and high-frequency, mostly expressed over the time window of the green Cu's diurnal

cycle ($\sim$11:00-18:00 EST). The different frequencies are related to different spatial scales, such that the lower the frequency is the larger is the spatial scale that dominates the corresponding EOF. The intermediate frequencies are further explored by focusing on an oscillatory pair–PC5 and PC6. These two components demonstrate a nearly-periodic $1.5\,hr$ oscillation with a phase shift, further indicating the existence of horizontally propagating GW, which appear as stripes in the corresponding EOFs.





The full reconstruction of the field (using 20 modes) highly resembles the "real" field (Fig. 6-(a,b)), eventhough the 20 modes explain only ∼65% of the variance. By inspecting the field's time–evolution through a transection along the GW (orthogonal to the cloud streets), we identify features like the diurnal cycle and the clouds streets (Fig. 6-(c)). However, the EOF decomposition allowed us to rebuild the field using sub-groups of modes, thus, separating between the scales (frequencies), and revealing different physical processes that interact to create the full complex patterns of the field. The reconstructed field's time-evolution

(along the transection) presents the cloud streets, the GW and the individual clouds in a much clearer and cleaner manner (Fig. 6-(d–f), respectively), allowing us to extract important parameters like the the cloud street's wavelength and aspect ratio and the GW period. Furthermore, the higher EOF modes reveal information regarding the organization of the field in higher order. Not only are the green Cu well organized as "pearls on a string" along the cloud streets, the clouds are also showing coherent spatial organization in the axis transverse to the streets. Clouds forming on adjacent streets tend to form on the same phase

and frequency, suggesting that a grid structure forms not only in the spatial domain (as was shown in Fig. 3), but also in the temporal domain, on the time scale of the cloud field's lifetime, indicating on a higher level of organization.

This study demonstrates that a standard decomposition of a shallow Cu field relying on EOFs, when performed on a well-prepared dataset, constitutes a useful tool for studying and characterizing the variety of patterns formed by the clouds within the field. The method allows indeed for disentangling the field's key organizational factors and for revealing hidden features

which otherwise would be hard or even impossible to distinguish from a naked eye analysis of the underlying satellite dataset. Dimensionality reduction when performed with modes that convey the right dynamical information, has proven its usefulness for the stochastic modeling and prediction of multiscale datasets in the recent years; see e.g. Kondrashov et al. (2018a, b); Chekroun et al. (2011). In the case of the dataset analyzed here, the EOF modes provide such an efficient reduction that allows for the identification and capture of key dynamical features (cloud streets and GW) and opens up, thus, new directions for

data-driven stochastic modeling of shallow Cu fields.

*Data availability.* All data used in this study are publicly available at: https://www.bou.class.noaa.gov/ (GOES–16 data), and https://worldview.earthdata.nasa.gov/ (topography data).

*Video supplement.* TEXT

*Author contributions.* M.D.C. conceived the presented idea. T.D. lead the analyses and M.D.C. supported. T.D., M.D.C., O.A., and I.K. dis-

cussed the results and wrote the manuscript. All co-authors provided critical feedback and helped shape the research, analysis and manuscript.

*Competing interests.* The authors declare that there are no competing interests.



*Acknowledgements.* This project received funding from the European Research Council (ERC) under the European Union's Horizon 2020 research and innovation programme (grant agreement No. 810370).



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
