# Peer review of "Deciphering Organization of GOES-16 Green Cumulus, through the EOF Lens"

_Atmospheric Chemistry and Physics, 2020_

## Author Comment (AC1)

**Reply to Reviewer #1**

*Manuscript ID acp-2020-1242*
*"Deciphering Organization of GOES–16 Green Cumulus, through the EOF Lens"*
*Tom Dror, Mickaël D. Chekroun, Orit Altaratz and Ilan Koren*
*ACP*
* * *
We are grateful for the careful report and positive comments of the reviewer about our manuscript and the pattern analysis preformed therein. In that respect, we enhance the visual rendering of the patterns shown in Fig. 2, 3 and 6. We hope that this paper will be helpful for the community, and will encourage the use of EOF and other decomposition methods in studying cloud field organization.

---

## Author Comment (AC2)

**Reply to Reviewer #2**

*Manuscript ID acp-2020-1242*
*"Deciphering Organization of GOES–16 Green Cumulus, through the EOF Lens"*
*Tom Dror, Mickaël D. Chekroun, Orit Altaratz and Ilan Koren*
*ACP*
* * *
We are grateful for the time and effort the reviewer invested in our work, and appreciate the careful reading of our manuscript. Below we address the reviewer's comments (our answer are marked in blue).
* * *
**Reviewer #2, comment #1**

**L45:** *are there any reasons why LES fail in reproducing shallow Cu, as LES can explicitly resolve shallow Cu at a common resolution of 100m.*

**Our response #2.1**

We thank the reviewer for this comment. While the resolution of LES is high enough to potentially resolve shallow Cu, they do not reproduce the organization patterns of shallow Cu. Most of the LES-based studies do not attempt to explore the patterns these clouds produce, and are aimed towards exploring other features of continental shallow Cu, namely, their diurnal cycle, cloud sizes and vertical extent, etc. Thus, the domain sizes used in these types of works are typically at the order of a few 10s of kilometers. These domain sizes are too small to reveal the mesoscale patterns described in the manuscript as from satellite observations. We believe that given larger domain sizes, and the right configuration, LES should be able capture organization patterns of shallow Cu, but this remains to be investigated.

**Reviewer #2, comment #2**

**L100:** *the identification of cloud objects, and the definition of cloud fraction and number of clouds are not clearly stated in the paper. I suggest the authors put more words on this part. In Figure 3c, it seems that the cloud fraction and the number of clouds is inconsistent.*

**Our response #2.2**

We thank the reviewer for this comment which helped making this point more precise. We have added more detailed descriptions into the definitions of the organization metrics introduced in **section 2.2.: L98–L104:**

"To derive the data driven organization metrics of Sec. 3, a threshold of 0.1 was applied to the **absolute** reflectance values to roughly discriminate between cloudy $(> 0.1)$ and non-cloudy $(\leqslant 0.1)$ **pixels. Cloud objects were then defined based on a pixel connectivity of four; cloudy pixels belong to the same cloud object if their edges touch, but not if their corners touch.** Metrics such as cloud fraction (CF), number of clouds (N) and the observed nearest neighbour cumulative density function (NNCDF) were calculated **for each image. Here CF is the sum of cloudy pixels over the sum of all pixels, N is the sum of all detected cloud objects, and NNCDF is the cumulative density function of the distance of each cloud object's centroid from its nearest neighbor.**"

Figure 3-(c) shows a boxplot of the CF and N for subdomain–1 (red) and subdomain–2 (blue) for the afternoon (15:02–18:02, when GWs are mostly pronounced). Subdomain–2 features a more evident signal of GWs in the afternoon, which causes the clouds there to become larger and more clustered. This signal is manifested in an increased CF, and a decreased N in subdomain–2 versus subdomain–1 (compare higher median of the blue box in the left side of panel (c), and lower median of the blue box at the right side of panel (c)).

**Reviewer #2, comment #3**

**L164:** *what's the criteria in judging an EOF mode degenerate or nondegenerate?*

**Our response #2.3**

Figure 4-(a) shows the the spectrum of the covariance matrix $C_X$, as composed of the eigenvalues $\lambda_1, \cdots, \lambda_{20}$. This spectrum informs us on the distribution of energy and on the separation/degeneracy of the EOF patterns. The degeneracy of an EOF mode is determined according to how separated its corresponding eigenvalue is from the others. A mode is said to be nondegenarate if this separation is unambiguous. This is what is observed for EOF mode 1 whose eigenvalue $\lambda_1$ is well separated on the vertical axis from the other $\lambda_k$s, even when including the errorbars. On the other hand, $\lambda_5 - \lambda_{20}$, belong to the flat part of the spectrum of $C_X$ and overlap almost completely, i.e., they are degenerate.

We added a clarification in section 4, L159–L163: "The spectrum of the covariance matrix $C_X$, shown in Fig. 4-(a), is composed of the eigenvalues $\lambda_1, \cdots, \lambda_{20}$ which informs on the distribution of energy and on the separation/degeneracy of the EOF patterns. The corresponding PCs are shown in Fig. 5-(a). The leading mode (EOF1, 31.8%) is nondegenerate (i.e., $\lambda_1$ is well separated from the rest the $\lambda_k$s, **see Fig. 4-(a)**) and explains more than twice the variance than any other mode."

**Reviewer #2, comment #4**

*L197: why PC5 and PC6 instead of other PCs can be used to identify GW. Are there any objective standards?*

**Our response #2.4**

We thank the reviewer for pointing this out. The PCs are here, to a first order, sorted according to their temporal frequency, such that higher PCs correspond to higher frequencies, i.e., faster changes in time. In the paper we show that there exist an intermediate range of frequencies (PC3–PC9), that matches the time-variability of gravity waves (GWs). This is first illustrated by the nearly oscillatory behaviour as shown for PC5–PC6 (Fig. 5). The reviewer is right in the selection of PC5–PC6 being somewhat subjective, among the aforementioned range of intermediate frequencies. However, in a second stage we use all the range of intermediate frequencies (PC3–PC9) to reconstruct the field, that help reveal the GWs signal by adopting a Hovmöller representation (see Fig. 6-(e)).

**Reviewer #2, comment #5**

*L255: Grammar check "such that the lower the frequency is the larger is the spatial scale that dominates the corresponding EOF".*

**Our response #2.5**

We thank the reviewer for this comment.

We changed the following sentence to L253-L254: "such that lower frequencies correspond to EOF modes that are dominated by larger spatial structures".